# Interior and Evolution of the Giant Planets

Yamila Miguel [1,2,*] and Allona Vazan [3]

1 Leiden Observatory, University of Leiden, Niels Bohrweg 2, 2333 CA Leiden, The Netherlands
2 SRON Netherlands Institute for Space Research, Niels Bohrweg 4, 2333 CA Leiden, The Netherlands
3 Astrophysics Research Center of the Open University (ARCO), The Open University of Israel, Raanana 4353701, Israel
* Correspondence: ymiguel@strw.leidenuniv.nl

**Abstract:** The giant planets were the first to form and hold the key to unveiling the solar system's formation history in their interiors and atmospheres. Furthermore, the unique conditions present in the interiors of the giant planets make them natural laboratories for exploring different elements under extreme conditions. We are at a unique time to study these planets. The missions Juno to Jupiter and Cassini to Saturn have provided invaluable information to reveal their interiors like never before, including extremely accurate gravity data, atmospheric abundances and magnetic field measurements that revolutionised our knowledge of their interior structures. At the same time, new laboratory experiments and modelling efforts also improved, and statistical analysis of these planets is now possible to explore all the different conditions that shape their interiors. We review the interior structure of Jupiter, Saturn, Uranus and Neptune, including the need for inhomogeneous structures to explain the data, the problems unsolved and the effect that advances in our understanding of their internal structure have on their formation and evolution.

**Keywords:** giant planets interiors; giant planets evolution; planet formation

## 1. Introduction

The giant planets' interiors are a window into the solar system's history. They formed before the gas dissipated from the primordial disk that gave birth to the sun—in a few millions of years [1]—saving in their interiors crucial information to understand the first stages of the formation of the solar system. Furthermore, Jupiter, Saturn, Uranus and Neptune contain 99.55% of the mass of all the planets, dominating the dynamics of the rest of the bodies and effectively determining the architecture of the solar system [2–5]. Studying the interiors of the giant planets also pushes us to study the properties of hydrogen, helium and mixtures of ice and rocks under extreme conditions [6,7], providing a natural laboratory for the study of elements at high pressures and temperatures.

We are at an incredible time to study the gas giants in our solar system. The Cassini and Juno missions have revolutionised our view of the interiors of Saturn and Jupiter, respectively [8–10]. Among the remarkable breakthroughs, we discovered that the interiors of the giants are not homogeneous [11–13], we revealed the extent of their differential rotation [14–16] and exposed the details of their magnetic fields [17]. Moreover, ground facilities [18], as well as the James Webb Space Telescope, will help to refine our knowledge of their atmospheres and aid in interior structure calculations. On the other hand, studies on Uranus and Neptune show the need for a future mission to these planets to break degeneracies and better determine their interior structure [19–21]. In this paper, we review our current knowledge of the interiors of Jupiter, Saturn, Uranus and Neptune, with a particular focus on the new data and the necessity of inhomogeneous structures to explain it and discuss how this reflects in our understanding of the giant planets' formation and evolution.

This paper is organised as follows. The fundamental properties of the giant planets and observational data used for interior structure and evolution calculations are shown

in Section 2. Section 3 summarises the state-of-the-art modelling efforts to understand these planets. Section 4 describes the current understanding of the interior structure of Jupiter and Saturn (Section 4.1) and Uranus and Neptune (Section 4.2). Section 5 describes the implications of the internal structure for our understanding of planet formation and evolution. Finally, Section 6 contains the summary and future prospects.

## 2. Constraints on the Giant Planets from Observations

Remote sensing and ground observations of the giant planets and their satellites have provided remarkable data on these planets. In this section, we discuss the most important observational constraints that are used to retrieve their interior structure and evolution.

In Figure 1, we show some of the fundamental properties of the four giants. While the mass of the giant planets is obtained by the measurements of the motions of their natural satellites [22–25], their radius is constrained by radio-occultation experiments [26–30]. In Figure 1, we also include a description of the planets' internal luminosity [31], which is an important constraint for evolution calculations.

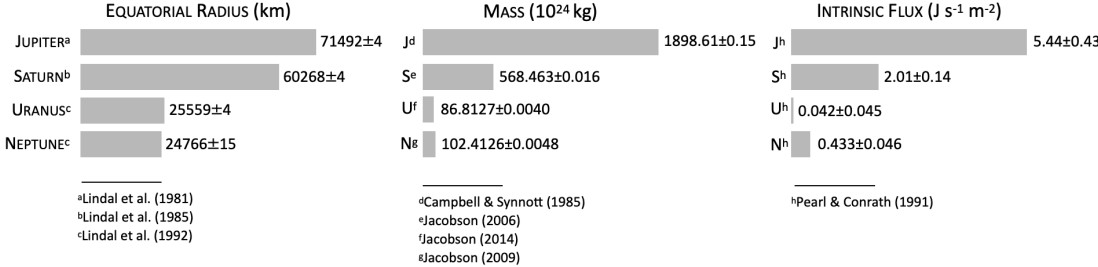

**Figure 1.** Fundamental properties of the four giants of the solar system [22–29,31].

### 2.1. Gravity Data

Besides the masses and the radii, we can use the planets' gravity fields as constraints to derive their internal structures. Because the giant planets rotate very fast, their fluid bodies deviate from a spherical symmetry and we can express their gravitational potential, $U(r, \theta)$, by an expansion in Legendre polynomials, $P_i(\cos \theta)$,

$$U(r, \theta) = \frac{G\,M}{r} \left[ 1 - \sum_{i=1}^{\infty} \left( \frac{R_{eq}}{r} \right)^{2i} J_{2i} P_{2i}(\cos \theta) \right] \tag{1}$$

where $G$ is the gravitational constant, $M$ is the planet's mass, $R_{eq}$ is its equatorial radius, and $J_{2i}$ are the gravitational harmonics. Equation (1) includes only even gravitational harmonics because it is assumed that the perturbation comes from the rotation of an axially and hemispherically symmetric giant planet. If we equate Equation (1) to the gravitational potential evaluated from the density distribution, we find the following expression for the gravitational harmonics:

$$J_{2i} = -\frac{1}{MR_{eq}^{2i}} \int \rho(r, \theta) r^{2i} P_{2i}(\cos \theta) d^3 r \tag{2}$$

where $\rho$ is the density. We can see from the dependence on $r^{2i}$ in Equation (2) and also from the schematic plot in Figure 2, that the higher-order gravitational harmonics contribute mostly to our understanding of the giant's external layers, while the low-order gravitational harmonics help us to constrain the giant planets' deeper interiors.

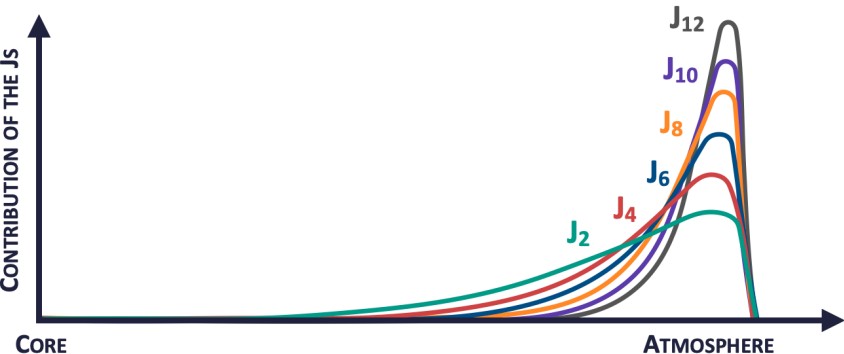

**Figure 2.** Schematic figure showing the contribution of the gravitational harmonics from the different regions in the interior of the planet. The low-order gravitational harmonics contribute to our understanding of giant planets' deep interiors and the high-order gravitational harmonics help to unveil deep atmospheric dynamics.

Figure 2 also shows that gravitational harmonics do not provide direct evidence to study the core of the giant planets, whose presence is derived indirectly through the constraints on outer layers (see Section 4).

The gravitational harmonics are inferred from the Doppler tracking data of a spacecraft during a flyby. The accuracy of the gravitational harmonics depends on the number of flybys and orbit of the spacecraft. With information from flybys by the missions Pioneer 10 and 11, Voyager 1 and 2, Cassini, New Horizons and two orbiters—the Galileo mission and the Juno mission—Jupiter is the giant planet for which we have more accurate gravity data. In particular, the Juno mission has been orbiting Jupiter since 2016, and its unprecedented close-in polar orbits have provided gravitational data at least an order of magnitude more accurate than all the previous measurements [8].

For Saturn, the most accurate gravity data to date are due to the close-in orbits of the Cassini mission during its last year. The so-called Cassini Grand Finale put the spacecraft in an unprecedented close orbit, allowing a more accurate determination of Saturn's gravity field [10].

With only one flyby by the Voyager 2, Uranus and Neptune have less accurate gravitational constraints, which translates to a larger uncertainty in the determination of their internal structure [32]. Figure 3 compares the gravitational harmonics $J_2$ and $J_4$ for the four giants, where we see that Jupiter and Saturn are the planets with more accurate gravitational data.

Odd Gravitational Harmonics and Differential Rotation

Odd gravity harmonics are zero for a giant planet that is axially and hemispherically symmetric. The giant planets are not perfectly symmetrical, though this was a good approximation, given the sensitivity of the measurements we had before the Juno and Cassini Grand Finale missions. Nevertheless, the remarkable data provided by these missions have allowed us to measure, for the first time, inhomogeneities in the mass distribution between the two hemispheres of Jupiter and Saturn, i.e., the odd gravity harmonics [9,10,15]. Table 1 shows the gravitational harmonics measured for the four giant planets. We see that Jupiter and Saturn have more gravitational harmonics observed than Uranus and Neptune, including odd gravitational harmonics.

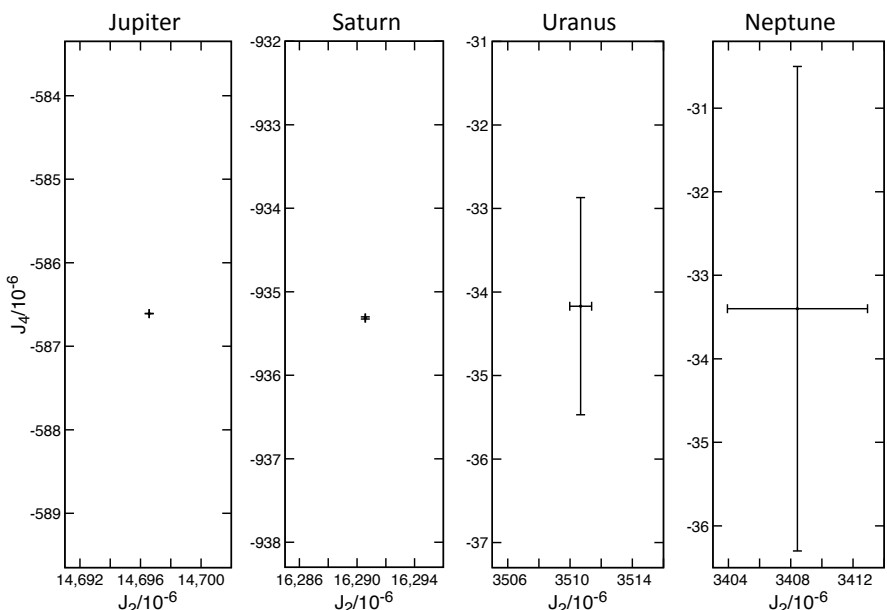

**Figure 3.** Gravitational harmonics J$_2$ vs. J$_4$ for all the giant planets in the solar system (see Table 1 for references). Different figures show the same range in the parameter space to compare the extent of the error bars among the planets.

Because we expect Jupiter and Saturn to rotate as rigid bodies in their deep interiors [14], the asymmetries observed are caused by atmospheric dynamics, providing a unique way of determining the extent of the differential rotation in these planets [33]. Using an approximation based on the thermal wind equation, Kaspi et al. [15] determined that the differential rotation extends up to 3000 km for Jupiter, an estimation that was further validated by independent calculations using the even gravitational harmonics by Guillot et al. [14]. Similarly, it was found that the extent of differential rotation in Saturn extends up to 9000 km [16]. In both cases, it is believed that the interaction of the material in the semiconducting region with the magnetic field leads to a decay of the flows [34]. While using less accurate measurements, some estimations were made for the depth of the zonal flows and extent of differential rotation for Uranus and Neptune, which was found to extend up to approximately 0.93–0.95 of the planets' radii using their gravity constraints [35] and through the analysis of the total induced Ohmic dissipation in their deep atmospheres [36,37].

**Table 1.** Gravitational harmonics of the giant planets.

| J$_n$ ($\times \mathbf{10^6}$) | Jupiter | Saturn | Uranus | Neptune |
|---|---|---|---|---|
| J$_2$ | 14,696.5735 ± 0.00056 [a] | 16,290.573 ± 0.0093 [b] | 3510.68 ± 0.70 [c] | 3408.43 ± 4.50 [d] |
| J$_3$ | −0.0450 ± 0.0011 [a] | 0.059 ± 0.0076 [b] | – | – |
| J$_4$ | −586.6085 ± 0.0008 [a] | −935.314 ± 0.0123 [b] | −34.17 ± 1.30 [c] | −33.40 ± 2.90 [d] |
| J$_5$ | −0.0723 ± 0.0014 [a] | −0.224 ± 0.018 [b] | – | – |
| J$_6$ | 34.2007 ± 0.00223 [a] | 86.340 ± 0.029 [b] | – | – |
| J$_7$ | 0.120 ± 0.004 [a] | 0.108 ± 0.0406 [b] | – | – |
| J$_8$ | −2.422 ± 0.007 [a] | −14.624 ± 0.0683 [b] | – | – |
| J$_9$ | −0.113 ± 0.012 [a] | 0.369 ± 0.086 [b] | – | – |
| J$_{10}$ | 0.181 ± 0.0216 [a] | 4.672 ± 0.14 [b] | – | – |
| J$_{11}$ | 0.016 ± 0.037 [a],* | −0.317 ± 0.1526 [b] | – | – |
| J$_{12}$ | 0.062 ± 0.0633 [a],* | −0.997 ± 0.224 [b] | – | – |
| Reference radius (km) | 71,492 [a] | 60,330 [b] | 25,559 [c] | 25,225 [d] |

[a] Durante et al. [38], see also Iess et al. [9]. * Note that uncertainties for J$_{11}$ and J$_{12}$ for Jupiter are larger than the measured values. [b] Iess et al. [10]. [c] Jacobson [25]. [d] Jacobson [24].

### 2.2. Atmospheric Abundances

An important constraint for planet formation and interior model calculations is atmospheric abundance. Atmospheric abundances are determined by disequilibrium processes such as vertical mixing, cloud and haze formation and photochemistry in the atmosphere and can be different than abundances found in the deeper interiors of the planets. Nevertheless, if the mixing is too strong, species can be carried out from the deeper convective region (the interior) to the atmospheric radiative zone, where they are observed [39,40]. Therefore, while they are not a direct representation of the entire bulk metallicity of the planets—because the planets have other layers in the interior and/or a gradient of heavy elements in some regions (see Sections 4.1 and 4.2)—they provide a boundary condition to be matched with interior model calculations in the outermost layers of the planet interior structure. Atmospheric abundances can also be key to constraint planetary formation models since they might carry some information on the feeding location of the giant planets during their accretion phase [41–46].

Observations of the chemical abundances in the giant planets' atmospheres were made using remote sensing from space and ground-based observations. Similarly to what we found for the gravity data, Jupiter and Saturn are the giant planets for which we have more detailed information of their atmospheres, while Uranus and Neptune are more uncertain. The most abundant elements in the giant planets' atmospheres are hydrogen and helium, which are some of the most difficult ones to detect. For Jupiter, helium abundances were provided by the Galileo probe that entered Jupiter's atmosphere and made in situ measurements, and is $0.82 \pm 0.02$ times solar [47,48], where we used the protosolar abundance reported by [49] as the solar value. Helium was not directly measured from in situ observations for the other giants but obtained via indirect analysis. In the case of Saturn, estimations were made based on the analysis of Cassini CIRS data and gave a helium abundance of $0.31 \pm 0.092$ solar [39,50,51]. For Uranus, the estimation us of $0.94 \pm 0.16$ solar [52,53] and for Neptune, there are two potential interpretations to explain the detection of HCL: one including $N_2$ that derives a helium abundance of $0.94 \pm 0.16$ solar and another one without $N_2$ in Neptune's atmosphere that obtains a helium abundance of $1.26 \pm 0.21$ solar [39,53,54].

The atmospheric abundance of the remaining heavier elements compared to hydrogen, or atmospheric metallicity, is obtained by using measurements of the most abundant heavy elements in these atmospheres. While oxygen is expected to be more abundant in the protosolar nebula and a better proxy of the metals in the planet's atmospheres, the abundance of oxygen is measured using the most abundant molecule bearing this element, which is water and is not so easy to observe. The giant planets in the solar system are cold, and water condenses, which makes measurements challenging. In situ observations by the Galileo probe and the microwave radiometer on board the Juno spacecraft were able to measure water abundance on Jupiter's atmosphere, giving an abundance between 1 and 5 times solar [55]. For Saturn, Uranus and Neptune, we do not have measurements of water abundance and use carbon abundance as a proxy of their atmospheric metallicity. These provide enrichments of $8.98 \pm 0.34$ for Saturn [51] and an estimate of $80 \pm 20$ for Uranus [56] and Neptune [57–59].

### 2.3. Atmospheric Temperatures

The limit between the atmosphere and the interior for the giant planets is difficult to determine. Traditionally, interior model calculations start at a pressure of 1 bar for the giants in the solar system. The entropy at this location, or, equivalently, the temperature at 1 bar, determines the initial adiabat for interior structure models. The temperature at 1 bar is usually measured using either radio occultation experiments or, in the case of Jupiter, using also in situ estimations. Nevertheless, a note of caution is that all these measurements provide local estimations of the temperature, and it is unclear how representative of the mean values in the entire planet they are. An example of this is Jupiter, where in situ measurements of the Galileo probe in one of Jupiter's dry spots located at a latitude of

6.57°N gave a temperature at 1 bar of $T_{1bar}$ = 166.1 ± 0.8 K [60], while a recent reassessment of the Voyager 1 radio occultation measurements gave a value of $T_{1bar}$ = 170.3 ± 3.8 K at 12°S (ingress) and $T_{1bar}$ = 167.3 ± 3.8 K at 0°N (egress) [30]. For Saturn, Voyager occultations give $T_{1bar}$ = 135 ± 5 K [27] and radio occultations with Voyager 2 indicate that Uranus and Neptune have values of $T_{1bar}$ = 76 ± 2 K [28] and $T_{1bar}$ = 72 ± 2 K [28], respectively.

*2.4. Magnetic Fields*

The magnetic fields of the giant planets are remarkably different, each one from the other, providing further constraints to their interior structures. For Jupiter, the magnetometer on board the Juno mission has shown that the magnetic field has a complex structure with most of the flux coming out from a narrow band located in the northern hemisphere and where much of it returns through a small and isolated flux spot called the blue spot, located in the same northern hemisphere close to the equator [17]. This non-dipolar nature of the magnetic field is located mostly in the northern hemisphere. The southern hemisphere is predominantly dipolar. This indicates that Jupiter's magnetic field is not generated in a unique homogeneous shell, but is a product of radial variations in the interior of Jupiter that could be due to the presence of different layers with differences in composition, electrical conductivity, or both [17,61,62].

In contrast to the complex structure observed in Jupiter, the Cassini Grand Finale magnetometer dataset has revealed that Saturn's magnetic field is dipole-dominated and has an almost perfect alignment with the planet's rotation axis, with a deviation of only 0.007° [63]. One of the interpretations of this highly axisymmetrical field is the presence of a stably stratified layer above the dynamo region, where a mechanism such as double-diffusive convection is acting, which will filter any non-axisymmetric internal magnetic field. The last estimations show that this layer could have a thickness of 2500 km [63,64], although this could be in disagreement with the internal constraints derived from the ring seismology [12].

The magnetic fields of Uranus and Neptune are weakly constrained by measurements by Voyager 2 that showed complex multipolar magnetic fields [65,66]. Given the poorly constrained data and the unknowns in the interior structure of Uranus and Neptune, these observations have different interpretations, and models struggle to explain the observations [67]. Some interpretations are consistent with the fields resulting from the thin-shell dynamo surrounding a stably stratified region [68], or with multipolar dynamos, the consequence of three-dimensional turbulence that may be excited in planetary-scale water layers [69].

## 3. Modelling of Giant Planet Interior and Evolution
*3.1. Basic Equations*

The models of the interior structure of the giant planets assume hydrostatic equilibrium, energy transport and mass and energy conservation:

$$\frac{\partial P}{\partial r} = -\rho g \tag{3}$$

$$\frac{\partial T}{\partial r} = \frac{\partial P}{\partial r}\frac{T}{P}\nabla_T \tag{4}$$

$$\frac{\partial m}{\partial r} = 4\pi r^2 \rho \tag{5}$$

$$\frac{\partial L}{\partial r} = 4\pi r^2 \rho T \frac{\partial S}{\partial t} \tag{6}$$

where $P$ is the pressure, $g = Gm/r^2$ is the gravity acceleration, $T$ is the temperature, and the temperature gradient, $\nabla_T$, depends on the process that dominates the energy transport. $L$ is the local intrinsic luminosity, $S$ is the specific entropy and t is the time.

In adiabatic models, the entropy is uniform along each composition layer, and thus $\nabla_T$ is taken to be the adiabatic temperature gradient, where only the outermost layer of the planet is taken to be radiative. In non-adiabatic models, entropy is not necessarily uniform, and $\nabla_T$ can be adiabatic, radiative and/or conductive. The heat transport mechanism is then determined by a convection criterion—the Schwarzschild criterion, or the Ledoux convection criterion [70], which takes into account also the effect of material distribution on heat transport. Other heat transport mechanisms that can take place in regions with non-uniform material distribution inside the planet are layered-convection (double-diffusive convection), in regions that are stable for convection according to the Ledoux convection criterion and unstable according to the Schwarzschild criterion [71], and moist convection, in which composition phase transition takes place [72,73].

Unlike in the adiabatic model, in which composition layers are uniform, in non-adiabatic models, material transport is part of the thermal evolution. Heat transport by convection in composition gradients can mix the composition in the convective layer (convective mixing). If convection is vigorous enough, the process is ended in uniform composition distribution in the convective cell. Convective mixing is usually modelled as a diffusive–convective flux of particles:

$$\frac{\partial X_j}{\partial t} = \frac{\partial}{\partial r} F_j \tag{7}$$

where $X_j$ is the number fraction of the $j$th element, and $F_j$ is the material flux of this element, proportional to the mixing length and velocity of the convection (see [74] for more details). Therefore, ideally, non-adiabatic evolution models should take into account both heat and material transport self-consistently, adding Equation (7) to the set of Equation (3).

### 3.2. Equations of State

In addition to the structure equations described in Section 3.1, an equation of state is needed to calculate the internal structure of the giant planets. The unique conditions of pressure and temperature inside the giant planets make the calculation of an equation of state complex because there are interactions of atoms, molecules and ions in a fluid partially degenerate. Furthermore, there is a non-negligible amount of heavy elements (especially in the case of Uranus and Neptune), and studies of mixtures of ices and rocks together with hydrogen and helium are essential to have a complete picture of the problem. This is not an easy task, and the determination of the interior structures is subject to the accuracy with which we know the equations of state [13,75–77]. A common assumption made when modelling the interior structure of the giant planets is to couple the pure equations of state for the different components using the linear mixing rule, where the value of a certain extensive variable (e.g., entropy) for the mixture corresponds to the addition of the variable calculated for each individual component and weighted by their respective mass concentrations [78]. Therefore, we can estimate the equations of state of each one of these components separately and then obtain the value for the desired composition.

### 3.2.1. Hydrogen and Helium

Hydrogen and helium are the main components of Jupiter and Saturn and two of the main constituents of Uranus and Neptune. Therefore, the equations of state of these elements are crucial to obtain a proper determination of the interiors of the giant planets. Figure 4 shows the phase diagram of hydrogen. The orange region indicates the region of interest for the interior structure of the four giant planets. We see that hydrogen can be found as a molecular gas or a metallic fluid, with a smooth first-order transition between the two phases [79,80].

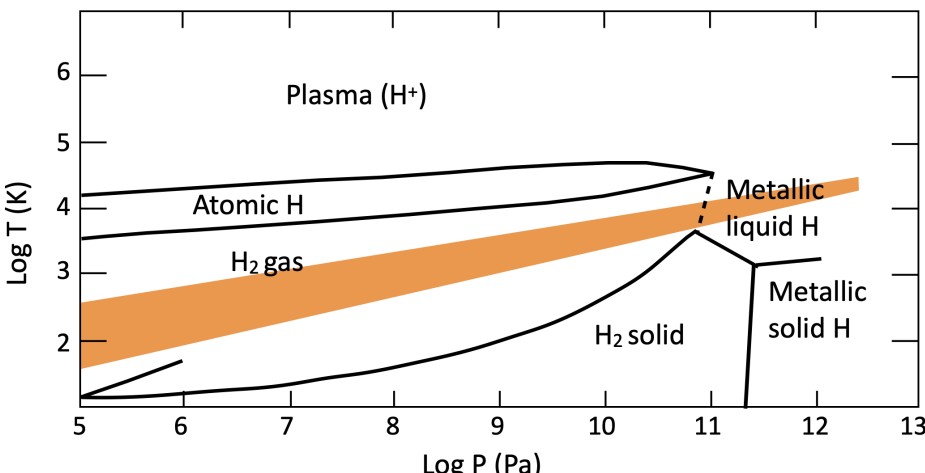

**Figure 4.** Phase diagram of hydrogen. The orange area represents the zone of interest for the interiors of the giant planets. Adapted from Miguel et al. [76].

The equation of state for hydrogen in the conditions that matter for giant planet interiors is calculated numerically through ab initio calculations using density functional theory [6,81], and compared with results from laboratory experiments performed with anvil cells and laser-driven shocks [82–84]. Nevertheless, while the experimental results are conducted at high pressures, they are not the same conditions of high pressures and low temperatures found in the interiors of the giant planets. Thus, we do not have direct confirmation of the equation of state calculated numerically at the conditions required to model the interiors of Jupiter, Saturn, Uranus and Neptune. As a result, there are some notable differences between the different equations of state published in the literature [6,76,85,86], where the interpolation between the different regimes plays an important role in shaping some of these differences. Another source of uncertainty when dealing with mixtures of hydrogen and helium comes from the use of the linear mixing rule, which adds an error coming from non-ideal effects that lead to differences in the equations of state [77] and result in remarkably different interior structures for Jupiter [13,76].

The interiors of the giant planets also have a non-negligible amount of helium, and equations of state for helium [87] and for a mixture of hydrogen and helium [6] are important to determine their interior structure. Some of the results found when studying the mixture of hydrogen and helium is that there is an immiscibility region that separates the solution of hydrogen and helium in two different components and affects the interior structures of Jupiter and Saturn [88–90]. This causes the separation of the envelopes of the two giants into two distinct layers separated by the immiscibility region. Due to this immiscibility, helium forms droplets that rain down, leading to a depletion of helium in the external envelope layer of Jupiter and Saturn and an overabundance in their interiors. This hydrogen–helium phase separation was theorized a long time ago [88,89] and received observational confirmation with the observed depletion of helium abundance in Jupiter's atmosphere by the Galileo probe [60]. Nevertheless, it was experimentally confirmed only recently [7], and many uncertainties still remain on this process and its effects on the interior structure of Jupiter and Saturn.

### 3.2.2. Ices and Rocks

The interiors of the giant planets also have a large amount of metals. While this is particularly important for the interiors of Uranus and Neptune, the relevance of the equation of state of mixtures of rocks and ices for the interiors of Jupiter and Saturn should not be neglected. Water is one of the most important ices in the outer solar system, and its relevance for the interiors of the giant planets has yet to be determined. Equations of state for water and mixtures of ices at the conditions found in the interiors of the giant planets

were published in the literature and show that water should be present as an ionic fluid in the interiors of Uranus and Neptune [91–93]. Furthermore, numerical calculations show that when mixing water with hydrogen and helium in the conditions found in the interiors of the giant planets, they are soluble and mix [94,95], allowing the presence of a dilute core in the interiors of Jupiter and Saturn [11] (see Section 4.1).

Rock-forming elements, such as mixtures of silicates and iron alleys, also need to be studied to obtain a complete picture of the interiors of the giant planets. A study by [96] shows that $MgSiO_3$ should separate and form $MgO$ and $SiO_2$ at the conditions found in the deep interiors of Jupiter and Saturn and should be in solid form at the lower pressures found in the interiors of Uranus and Neptune. When looking at the phase diagram of $MgO$, it is found that this element should be in solid form at the conditions found in the cores of Jupiter and Saturn [96], and the same was found for iron [96]. Nevertheless, ab initio calculations show that iron but also $MgO$ and even $SiO_2$ are expected to be soluble in metallic hydrogen, so it is not entirely clear if the planets should have a solid core made of pure rocks or a fluid mixture of different components. Moreover, water–rock miscibility at high pressure, as was recently found by laboratory experiments [97,98] and by numerical simulations [99,100], indicate that ice and rocks are probably mixed in the interiors of the giant planets (see Section 4.2 below for more details). More work needs to be done to understand the nature of these mixtures in the deep interiors of the giant planets.

## 4. Internal Structure of the Giant Planets

The observational constraints described in Section 2 are used together with the models detailed in Section 3 to recreate the interior structure of the giant planets. Some big questions that these models try to answer are the amount and distribution of the different elements in the planet's interiors. In particular, the amount and distribution of heavy elements are crucial constraints for planet formation models. The most accepted theory to explain the giant planet formation is the core-accretion mechanism, where planets first form a core through collisions of solids followed by the accretion of a gaseous envelope [101]. There is an ongoing discussion in the community regarding the solid accretion that may happen through the accretion of bigger planetesimals or small pebbles. These different theories might lead to a different distribution of heavy elements in the planet's interiors: either a high-enrichment (planetesimals) or low-enrichment (pebbles) of the envelope [102–106]. Figure 5 shows the result of some model calculations on the metallicity of the giant planets.

The figure shows that the total amount of metals increases when we go to smaller planets. It also shows a large dispersion in the results. Jupiter has an amount of metals that ranges between 18 and 45 $M_\oplus$ [13,107–109]. We note that smaller values, as small as 11 $M_\oplus$, were also found by [13,108,110] when using the equation of state by [85], which does not consider non-ideal mixing effects and, therefore, is not accurate enough to be used for the modelling of the giant planets in the solar system. The differences in the results between Jupiter models are caused by the use of different model assumptions, in particular, the equation of state for hydrogen. Hydrogen is the main component of Jupiter, and the model results are very dependent on the accuracy of the equation of state [13,75–77]. For Saturn, besides the gravity information obtained with remote sensing, we also have information provided by ring seismology. This is a novel technique that uses the effect that Saturn's oscillations have on its rings to study the interior properties of the planet [12,111–113]. The results of ring seismology analysis [12] and traditional interior modelling [10,114,115] inform us that Saturn has $19 \pm 1\,M_\oplus$ of metals in its interior. For Uranus and Neptune, the uncertainties in the observational constraints translate into uncertainties in the determination of their interior properties. These planets have more than 80% of metals in their interior. The mass of metals ranges between 11 and 13 $M_\oplus$ for Uranus and 13 and 15 $M_\oplus$ for Neptune, when using both adiabatic [116,117] and non-adiabatic structures [118] (see Section 4.2).

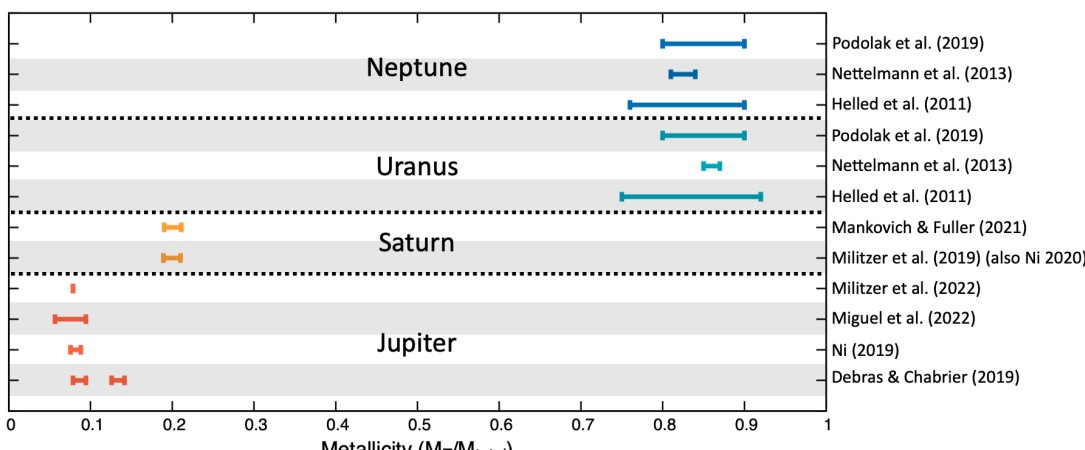

**Figure 5.** The mass of heavy elements divided by the planet's mass as obtained by different modelling efforts indicated in the y-axis. We note that the model by [107] has two different range of values corresponding to different model assumptions [12,13,107–109,114–118].

Besides the total mass of metals, their distribution in the interior of the planets is of crucial importance when understanding giant planet formation. Figure 6 shows a schematic view of our current understanding. We see that Jupiter and Saturn are clearly different from Uranus and Neptune, although each one of these planets have their own peculiarities as will be described in Sections 4.1 and 4.2.

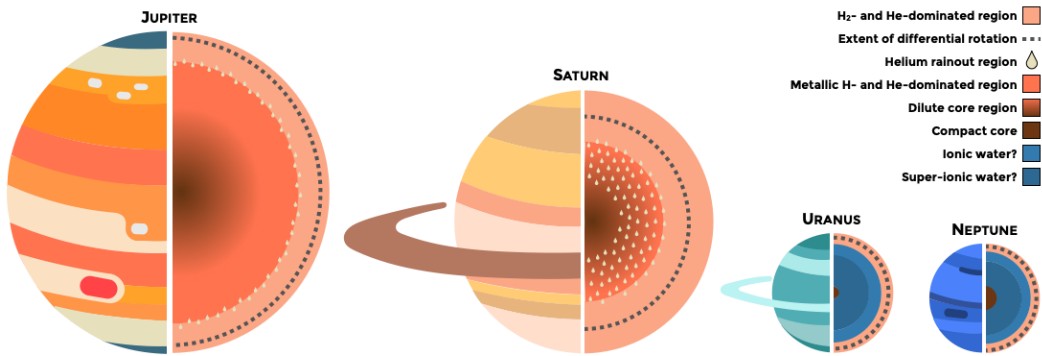

**Figure 6.** Schematic figure showing the interior structure and relative sizes of the four giants in the solar system. Note that current gravity measurements are not accurate enough to distinguish between different structures for Uranus and Neptune, but mixed rock and ice cores and inhomogeneous structures are also possible (see Section 4.2). Rings and other atmospheric features are not on scale.

### 4.1. Jupiter and Saturn: Dilute Cores and Non Adiabatic Structures

Jupiter and Saturn are mainly made of hydrogen and helium, so we need a deep understanding of these two elements and their mixtures to get a proper interpretation of the observed data. For many years, and previous to the arrival of Juno to Jupiter, models of Jupiter and Saturn divided the planets into three homogeneous, convective layers: a big compact core made of 100% heavy elements, and two envelope layers. These layers were an external one dominated by molecular hydrogen and depleted in helium, and an internal layer, where hydrogen was in a metallic form and helium was enriched compared to the protosun [76,119–121]. In this simple picture, the two envelope layers were divided at the location of the helium phase separation according to numerical estimations [90] and the core of the planets contained most of the mass of metals, reaching values of around 15 $M_\oplus$ for Jupiter [121] and 18 $M_\oplus$ for Saturn [114].

This picture changed dramatically with the arrival of the Juno mission at Jupiter and with the observations of the last year of the Cassini mission. For Jupiter, the extremely accurate gravity data (particularly $J_4$, $J_6$) and measurements of water abundance in the equatorial region of Jupiter's atmosphere [55] proved to be very difficult to reproduce by simple interior model calculations [8]. There are different explanations for the discrepancy between the model efforts and the observations: either the interior structure of the planets is more complex than the classical 3-layer picture, or our understanding of the equation of state of hydrogen, and hydrogen and helium mixtures was not complete [122]. While current models are still struggling to reproduce all observational constraints [13,107,109,110], it is clear that the right answer is probably a combination of the two reasons given above. The first models that got closer to reproducing the observational constraints provided by Juno were published by [11]. In these models, the authors brought back the idea that the giant planets might have a dilute core, an idea theorized in the 1970s [123], but they performed numerical calculations that introduced a dilute core in Jupiter's interior to reproduce Juno's gravitational measurements. The idea of a dilute core is that there is no sharp boundary between a core made of 100% heavy elements and the hydrogen and helium envelope above it, but rather that the core mixes and is diluted into the H-He envelope and a gradient of heavy elements is present in the deep interior of the planets (see Figure 6). A dilute core is needed to explain the measured $J_4$, and its presence in the interior of the giant planets is further supported by ab initio numerical calculations that showed that metals are miscible in metallic hydrogen at the pressures and temperatures in the dilute core region [94,95]. The presence of a dilute core also reinforces the concept that giant planets might not be made of simple convective layers as was previously thought, but instead have regions with a gradient of heavy elements in their interiors. Furthermore, in a recent paper by [13], the authors found that the two envelope layers in Jupiter also need to have an inhomogeneous distribution of heavy elements, with a larger enrichment in the inner layer [13]. The idea of inhomogeneous, non-adiabatic structures for Jupiter and Saturn was also discussed in the past, with a series of extreme but informative models by [124,125]. Nevertheless, current state-of-the-art models with dilute cores still struggle to reproduce the large abundance of metals in the atmospheres of these planets and have used different resources to find solutions that fit all the observational constraints, going from using higher temperatures at 1 bar [13,110], higher pressures for the helium immiscibility region and a larger abundance of metals in the outer envelope layer [107] or a modification of the observed zonal winds in the determination of the differential rotation of the planet [109]. Neither of these solutions is perfect. While a larger temperature at 1 bar was pointed by a recent re-analysis by the Voyager data [30], it is still a few degrees short of what is required by the interior models. Additionally, while recent laboratory data suggest that the pressure at the immiscibility region can be larger than determined by numerical experiments [7], there is still a discrepancy between the equations of state calculated numerically and the results found at very peculiar conditions in the laboratory. Furthermore, the possibility of a larger abundance of metals in the outer envelope layer could be a short-lived solution due to Rayleigh–Taylor instabilities. Finally, the idea of modifying the zonal winds to accommodate the differential rotation of the planet and be able to match the observations is not backed up by recent studies of the circulation of the deep atmospheres of Jupiter, which show that the meridional profile of the cloud-level wind extends to depth, giving no room for the modification of these [126,127]. These results show once more, that one of the most likely explanations of the discrepancy between the modelling efforts and the observations is due to either inaccuracy of the equation of state [13,77] or due to other physical mechanisms not currently considered in current state-of-the-art models. The new Juno data have raised the accuracy of the interior structure models for Jupiter and we now know that a dilute core is imperative to match the observations on the gravitational data. However, we also face further challenges that will be the focus of the modelling efforts in the coming years.

For Saturn, an exploration of Saturn's rings has revealed waves driven by pulsation modes within the planet that indicate the presence of a stably stratified layer in the deep interior of the planet, which can be caused by strong compositional gradients in its interior, indicating the presence of a dilute core [12] (Figure 6). Nevertheless, Saturn's dilute core would be very extended, going up to 60% of Saturn's radius, while Jupiter's dilute core could extend up to half the planet's radius and even less [11,13,77]. Moreover, the two giants also have other remarkable differences, such as the width of the helium rain region, which is expected to be more extended in Saturn, with unknown consequences for its interior structure. Some other differences between the planets that translate into differences in their interior structure are related to the magnetic fields, which are very axisymmetric in the case of Saturn and extremely in the case of Jupiter. Nevertheless, models that combine magnetic field and interior structure calculations remain to be done.

*4.2. Uranus and Neptune: Ice-Rock Mixtures and Non Adiabatic Interiors*

Uranus and Neptune, the so-called "ice giants", are expected to contain significant amounts of icy materials by virtue of their distance from the Sun. The traditional models of the interior structure of Uranus and Neptune, assume that the planets have a few layers: a compact core, an ice layer and a gas layer, with uniform composition in each one (see Figure 6). In these models, the heat transport in the layers is by convection, resulting in adiabatic interior structures. However, adiabatic models have difficulties explaining the observed luminosity of Uranus [117,128] and, although less significantly also of Neptune [129]. This means that the interiors of the two planets cannot be structured in a few uniform-composition layers, and alternative models are required to explain the measurements. Such an alternative is a model with a gradual distribution of metals in the interior, rather than with distinct layers of different compositions. Gradual composition distribution in the ice giants has been suggested to explain the observed luminosity of Uranus [116,130–132], and of Neptune [133]. Such a structure is also consistent with the complex magnetic fields of the two ice giants [68,134], as discussed in Section 2.4. Another alternative explanation for the anomalously low heat flow of Uranus is the presence of a frozen core [135]. Yet, the current measurements of Uranus and Neptune are poor, and consequently, the observations cannot distinguish between the suggested models, and there is a great degeneracy in terms of the shape of the composition gradients and their content. As of today, more observations are needed in order to achieve further progress in understanding the interiors of the two planets [136–138].

The lack of accurate measurements and the coverage of the gas envelope make it hard to figure out what fractions of the ices and rocks are in the interiors of Uranus and Neptune [139]. While observations of atmosphere-free Kuiper belt objects suggest an ice-to-rock ratio of 1:2 [140], disk chemistry predicts a larger ice fraction in the locations of Uranus and Neptune [141]. Moreover, the water ice content in the disk may vary with short-lived radiogenic heating at planet formation [142]. Thus, although called "ice giants", the ice content of Uranus and Neptune is still unknown, where most interior models lay between a 1:2 and 2:1 ice-to-rock ratio.

The distribution of the ice and the rock in the interiors of the ice giants is another mystery. While the simple layered structure contains a layer of ice on top of a rocky core, recent works show that ice and rock are expected to be miscible, and therefore mixed, at tens of GPa pressure [97–99,143] (see Figure 7). As a result, the ice and rock in the interiors of Uranus and Neptune are probably mixed, or at least partially mixed, and no distinct rock and ice layers are expected. Interior models with ice and rock mixtures are consistent with Uranus and Neptune measured properties [130–132,144]. Although hydrogen is not in its metallic form in Uranus and Neptune, water is miscible in hydrogen in the deep gas envelope of the ice giants, probably forming an additional gradient of composition, and affecting the interior structure and its evolution [144].

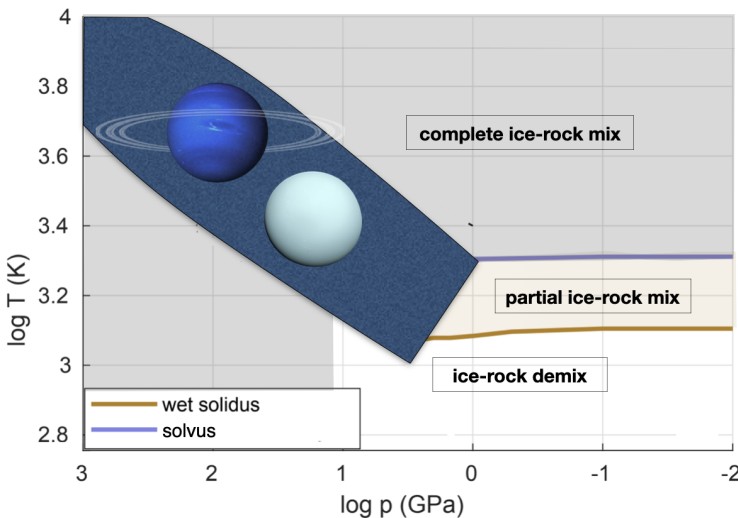

**Figure 7.** Ice–rock interaction regimes in pressure–temperature space, adopted from [143], with the area relevant for interiors of Uranus and Neptune marked in blue. The marked regime for Uranus and Neptune is the range of possible ice and rock distribution in Uranus and Neptune interiors, based on various structure and evolution models.

## 5. Implications for Planet Formation and Evolution

### 5.1. Planet Formation

Jupiter and Saturn contain much more gas than metals. This means that their formation happened early enough in the disk lifetime to allow for massive (runaway) gas accretion before the gas dissipated from the protoplanetary disk. Uranus and Neptune, by their size and location, formed slower and/or later. While Jupiter and Saturn went through the runaway gas accretion to become gas giant planets, Uranus and Neptune did not reach the gas accretion phase, and remained much less massive, with a much lower gas mass fraction (10–25% depends on the model, see Section 4). In this sense, Uranus and Neptune are stripped cores of gas giants. The pioneering work of Pollack [101] for Jupiter formation set the basic model for planet formation by Core-Accretion. Since then this model, and many improved versions of it including solid accretion by big, km-sized planetesimals [145], small, cm-sized pebbles [146], or both [106], were used to model giant planets, in our solar system and beyond. In these models, solids from the protoplanetary disk are accreted to form a planetary core, which stimulates the accretion of hydrogen and helium from the disk by the gravity of the growing core. The resulting interior is a solid core surrounded by a gas envelope. The assumption in these models is that all accreted solids reach the core, and all the gas is in the envelope, with no interaction between the core and the envelope.

However, some studies challenge this simplified core-envelope structure, showing that a substantial amount of the core building blocks (rocks, ices) dissipates in the growing envelope and does not reach the core [147–149]. As a result, the envelope is polluted with metals. Detailed planet formation models, which include the solid–gas interaction, show that the planetary interior at the end of the planet formation phase contains a gradual composition distribution, where metal mass fraction decreases gradually from the deep interior to the gas envelope [106,150–154]. Figure 8 shows an example of the primordial distribution of heavy elements found in these models, where inhomogeneous structures are a natural result of the formation models. In addition to planet formation models of Jupiter [122,150,155], calculations done for Uranus and Neptune [156] also show that composition gradient is a natural result of core accretion planet formation by either pebble or planetesimal accretion.

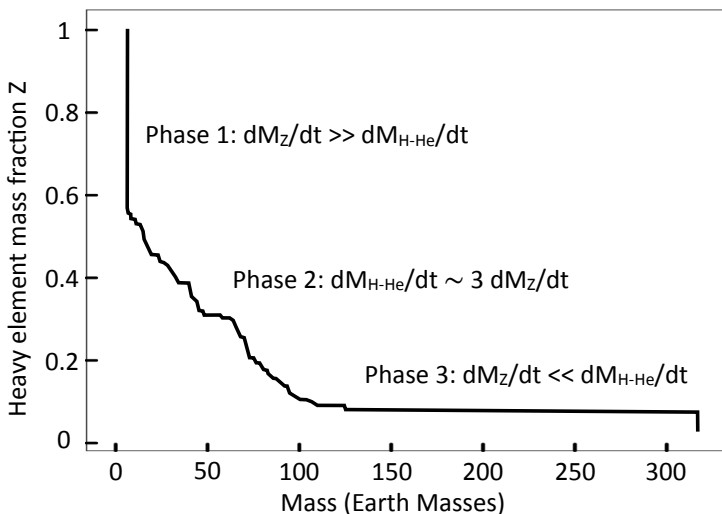

**Figure 8.** Distribution of the heavy-element mass fraction in Jupiter's interior obtained by formation models. The model assumes that Jupiter is formed in different stages and the core is formed by pebble accretion followed by planetesimal accretion and finally rapid gas accretion [106,152,153]. Figure adapted from [45].

The other theory of giant planet formation is gravitational instability, where giant planets are formed by the collapse of a gas clump in the disk. However, in the solar system, this mechanism is less likely and has severe difficulties to explain the formation of the four giant planets in our solar system [157].

*5.2. Evolution*

After their formation, the giant planets went through a long 4.5 Gyr phase of evolution to the stage we observe them today. During this period, the planets cooled by luminosity, and their interiors evolved accordingly. Thermal evolution modelling is the way to track the history of each planet to its current stage.

Thermal evolution models of the giant planets with composition gradients are conceptually different from the thermal evolution of interiors with a few homogeneous layers. The cooling of the latter is by large-scale convection, and the models are adiabatic structure models, where only the outermost thin layer of the envelope is radiative. The thermal evolution of an adiabatic planet is based on uniform entropy layers, where cooling is limited by the radiative outer envelope. In this model, the cooling is uniform and efficient (by convection). However, evolution models for the giant planets, which do not consider that non-adiabatic interior effects fail to fit the observed properties of the giant planets. The adiabatic models cannot fit properties such as the high luminosity of Saturn [123,158], the low luminosity of Uranus [117,128,159], and the luminosity of Neptune [129] at the age of the solar system.

In the alternative picture, where the interior contains gradual composition distribution, the heat transport does not operate only by large scale (adiabatic) convection, and thus is called non-adiabatic (actually, a more accurate term would be "not-fully-adiabatic" since parts of the planets can still be convective and thus adiabatic in many cases, yet the shorter term "non-adiabatic" is used for simplicity). The non-adiabatic evolution is usually slower and less efficient than the adiabatic cooling, caused by the effect of composition distribution on heat transport. The existence of a (stable) composition gradient suppresses large-scale convection (see Section 3.1), acting as a thermal boundary, slowing the cooling of the region interior to the gradient. In general, the non-adiabatic evolution leads to higher temperatures in the deep interior and super-adiabatic thermal profiles.

Non-adiabatic models, caused by a more realistic distribution of heavy elements in the interior, influence the observed properties of the giant planets. Thermal evolu-

tion models with composition gradients successfully explain the observed properties of the giant planets in our solar system, both the gas giants [122,125,160–162] and the ice giants [118,132,133,163]. The higher (than adiabatic) temperatures in the interiors of the giant planets influence the prediction of the heavy element mass fraction, as higher temperatures require more metals to fit the mass–radius relation at present [74,118,124].

Moreover, the non-uniform composition distribution can change along the evolution. Two main material transport mechanisms are convective mixing (upwards process), and settling (downwards process). Convective mixing (advection) takes place when large-scale convection is generated at regions with gradual composition distribution. In this scenario, the large-scale convection acts to smear the composition gradient and homogenise the convection cell [164]. In Jupiter, the mixing by convection is suggested to enrich the outer envelope with metals as the planet evolves [160,162,165] and takes place only in the first 1 Gyr of its evolution [162]. Evolution models for planets with composition gradients can be used to constrain the initial energy content of the planets, whereas models with too-hot initial interiors fail to fit Jupiter's observations [165].

Composition gradients in the interiors of Uranus and Neptune are expected to be steeper than those in Jupiter and Saturn [118,132,138], and are more stable against convective mixing. In this context, the composition of the gas envelopes of the ice giants is more primordial. Yet, the initial energy content can be constrained from the same argument, where Uranus was found to contain at most 20% of its accretion energy at the end of the planet formation phase [132] (Figure 9).

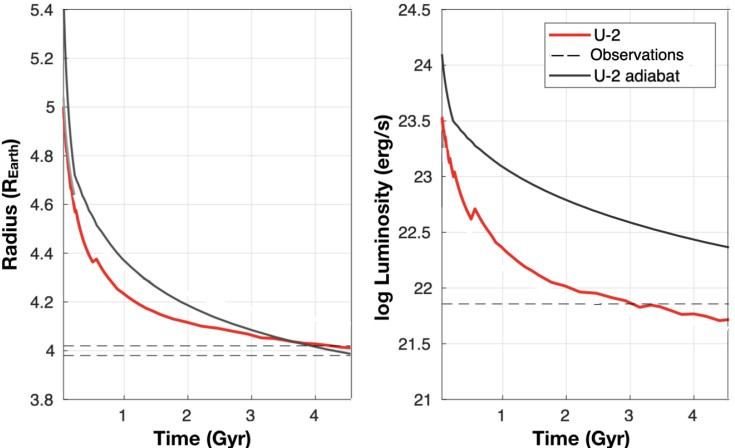

**Figure 9.** Effect of non-adiabatic evolution on observed radius and luminosity of Uranus: radius (**left**) and luminosity (**right**) of the non-adiabatic model (red) in comparison to adiabatic model (black). The horizontal dashed lines are the measured radius (range) and luminosity (upper bound). As can be seen, both models fit the observed radius, but the adiabatic evolution results in too high luminosity at present. The non-adiabatic model with self-consistent material transport fits both radius and luminosity observations. Figure adopted from [132].

The settling of condensates or immiscible elements is also contributing to interior evolution. One important settling process is the helium rain in the interiors of Saturn and Jupiter [166]. In this process, helium, under certain pressure–temperature conditions, separates from hydrogen and settles to deeper layers [123]. This process may lead to the formation of a helium-rich shell above the heavy-element deep interior [121,167] (Section 3.2). The energy released in the helium rain slightly increases the planet's luminosity. The helium-rich layer changes the interior structure and in particular the uniformity of the envelope. This in turn can change the heat transport from the deep interior, usually acting as an additional thermal boundary [160,161]. In the ice giants, water settling is the important settling process, either by condensation or by immiscibility. The settling of volatile species

in the planetary envelope affects the energy transport in the outer envelope of the ice giants and therefore may explain some of the differences in their luminosity [144,168,169].

## 6. Summary and Future Prospects

We are at an exceptional time to study the interiors of the giant planets. For Jupiter and Saturn, the Juno and Cassini missions have provided remarkable data that revolutionized our view of the interiors of these planets. Furthermore, we have more than 5000 exoplanets discovered, many of them giant planets, and the lessons learned from the detailed data and analysis of the giant planets in the solar system are now more important than ever [19].

Jupiter, Saturn, Uranus and Neptune have had an undeniable influence on the formation and evolution of the solar system. Therefore, a more detailed picture of the interior structure of these planets and the amount and distribution of the heavy elements inside them provides invaluable constraints to planet formation and evolution models. One of the biggest lessons learned is that giant planets do not have completely homogeneous structures, not their cores nor their envelopes, which directly translates into the formation mechanism. Jupiter and Saturn's dilute cores might have been a product of the formation mechanism and what is remaining of their primordial structures. Thermal evolution modified their interiors through time, allowing the creation of immiscibility regions and further separating the envelope into different parts, shaping the interior structures as we see them today. On the other hand, both numerical and laboratory experiments are showing that Uranus and Neptune might also have inhomogeneous interiors where the rocks and ices mix with the primordially accreted hydrogen and helium in their envelopes. These inhomogeneous structures in the four giants might have strong compositional gradients, which have the potential to affect the energy transport mechanism, modifying their temperature and luminosity, and changing the traditional way of modelling these planets.

These last years have been crucial and allowed exponential growth in our knowledge of these planets, but many challenges still remain. On the one hand, current state-of-the-art models of Jupiter and Saturn cannot explain all the observational constraints. A deeper knowledge of the equation of state and more detailed studies on other physical mechanisms in the interior of these planets are needed. On the other hand, with only one passage of Voyager 2, data on Uranus and Neptune are scarce, and more constraints are needed to break the degeneracies and have a deeper knowledge of their interior processes. New space missions for Uranus and Neptune are being proposed [20,170], which is the next step to move forward in this field. Finally, exoplanetary science is growing exponentially. New facilities, such as the JWST and future space (Plato, Ariel) and ground-based facilities (ELT), are pushing these studies further, allowing the detailed characterisation of these worlds and providing a large amount of data that will impact the perception of our solar system. Comparative planetology is the new frontier in giant planet research and a bright one to come.

**Author Contributions:** Y.M. wrote the parts relevant for planet interiors, particularly of Jupiter, Saturn and classical models for Uranus and Neptune. A.V. wrote the parts relevant for evolution of planets and non-traditional models. All authors have read and agreed to the published version of the manuscript.

**Funding:** This research received no external funding.

**Data Availability Statement:** This is a review paper and as such no new data was created. All data is taken from publications cited in the reference section.

**Conflicts of Interest:** The authors declare no conflict of interest.

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
