# Peer review of "Interior and Evolution of the Giant Planets"

_remotesensing, doi:10.3390/rs15030681_

Round 1

Reviewer 1 Report

This review paper is written clearly and very informative. The long list of references is appropriate. I recommend the pubblication in the present form; I just suggest to explain better the meaning of the Figure 2 (page 3 of 22) in the caption.

Author Response

We thank the referee for this comment. We included this suggestion in the new version of the manuscript. The caption of figure 2 now reads: "Schematic figure showing the contribution of the gravitational harmonics from the different regions in the interior of the planet. The low-order gravitational harmonics contribute to our understanding of giant planets' deep interiors and the high-order gravitational harmonics help to unveil deep atmospheric dynamics.”

Reviewer 2 Report

The authors have presented a careful and thorough review of the current studies concerning the interior and evolution of the giant planets. The work is very timely and I think it merits publication. But I have a few suggestions for the authors’ consideration.

1. Table 1 lists the gravitational harmonics of the giant planets. It is better to give the reference radii for the giant planets since the gravitational harmonics is dependent on the reference radius.

2. Lines 291-292: As we know, the amount and distribution of heavy elements are important for planet formation models. The authors should explain this in more details. It would be very instructive for unfamiliar readers.

3. Ni (2020) considered possible uncertainties in Saturn’s interior and evaluated the total amount of heavy elements in Saturn. It is better to illustrate their results in Figure 5 for comparison.

4. In Subsection 4.1, the authors emphasize the dilute core is needed to explain the gravity measurements. They should not forget that the existence of a dilute core poses a tough challenge for modeling planetary interiors: new observables are required to constrain the interior of the giant planets in addition to the gravity measurements.

5. There are some misprints which should be corrected. For example, the latitude should be replaced by 6.57 degree (line 156), and the word “co” should be replaced by “core” (line 409).

6. The authors should examine their citations carefully to avoid the wrong citation “?”.

7. The figures of this manuscript should be labeled with corrected numbers. Please see lines 426 and 457.

Author Response

The authors have presented a careful and thorough review of the current studies concerning the interior and evolution of the giant planets. The work is very timely and I think it merits publication. But I have a few suggestions for the authors’ consideration.
Table 1 lists the gravitational harmonics of the giant planets. It is better to give the reference radii for the giant planets since the gravitational harmonics is dependent on the reference radius.

Reply authors (RA): We thank the referee for this comment. We included now the reference radii for each giant planet in the last column of the table. 

2. Lines 291-292: As we know, the amount and distribution of heavy elements are important for planet formation models. The authors should explain this in more details. It would be very instructive for unfamiliar readers.

RA: The referee is right that explaining this more would be instructive to the reader. We added a few sentences in lines 292 to 300, that now read: “The most accepted theory to explain the giant planet formation is the core-accretion mechanism, where planets first form a core through collisions of solids followed by the accretion of a gaseous envelope \citep{Polack1996}. There is an ongoing discussion in the community regarding the solid accretion, that may happen through the accretion of bigger planetesimals or small pebbles. These different theories might lead to a different distribution of heavy elements in the planet's interiors: either a high-enrichment (planetesimals) or low-enrichment (pebbles) of the envelope \citep{Mousis2009, Lambrechts2012, Bitsch2015, Johansen2017, Alibert2018}.”

3. Ni (2020) considered possible uncertainties in Saturn’s interior and evaluated the total amount of heavy elements in Saturn. It is better to illustrate their results in Figure 5 for comparison.

RA: We thank the author for this remark. Because the Ni (2020) results are similar to those found by the other two authors (Mankovich & Fuller 2021, and particularly Militzer et al. 2019), we did not include them in the figure since they are not adding new information. Nevertheless, we added this reference in the figure and when talking about the metals in Saturn’s interior in the text. 

4. In Subsection 4.1, the authors emphasize the dilute core is needed to explain the gravity measurements. They should not forget that the existence of a dilute core poses a tough challenge for modeling planetary interiors: new observables are required to constrain the interior of the giant planets in addition to the gravity measurements.

RA: We thank the referee for this remark. The gravitational harmonics do help to constrain the dilute core, because this extends up half the radius of the planet, a dilute core is needed in order to explain the measurements of J4 and J6. We do agree with the referee that in order to explain the deep compact inner core we need more data, which is discussed in section 2.1 when describing the gravity data. In that section, we wrote: "Figure 2 also shows that gravitational harmonics do not provide direct evidence to study the core of the giant planets, whose presence is derived indirectly through the constraints on outer layers”.  

5. There are some misprints which should be corrected. For example, the latitude should be replaced by 6.57 degree (line 156), and the word “co” should be replaced by “core” (line 409).

RA: We corrected the typos. 

6. The authors should examine their citations carefully to avoid the wrong citation “?”.

RA: The misplaced references have been corrected. 

7. The figures of this manuscript should be labeled with corrected numbers. Please see lines 426 and 457.

RA: The labels have been corrected.